# The Predictive Value of miR-16, -29a and -134 for Early Identification of Gestational Diabetes: A Nested Analysis of the DALI Cohort

**DOI:** 10.3390/cells10010170

**Published:** 2021-01-15

**Authors:** Anja Elaine Sørensen, Mireille N.M. van Poppel, Gernot Desoye, Peter Damm, David Simmons, Dorte Møller Jensen, Louise Torp Dalgaard

**Affiliations:** 1Department of Science and Environment, Roskilde University, 4000 Roskilde, Denmark; ltd@ruc.dk; 2Institute of Human Movement Science, Sport and Health, University of Graz, 8010 Graz, Austria; mireille.van-poppel@uni-graz.at; 3Department of Obstetrics and Gynecology, Medizinische Universitaet Graz, 8036 Graz, Austria; gernot.desoye@medunigraz.at; 4Center for Pregnant Women with Diabetes, Department of Obstetrics, Rigshospitalet, Department of Clinical Medicine, University of Copenhagen, 2100 Copenhagen, Denmark; pdamm@dadlnet.dk; 5Macarthur Clinical School, Western Sydney University, Sydney 2560, Australia; Da.Simmons@westernsydney.edu.au; 6Steno Diabetes Center Odense, Department of Gynecology and Obstetrics, Odense University Hospital, 5000 Odense, Denmark; Dorte.Moeller.Jensen@rsyd.dk; 7Department of Gynecology and Obstetrics, Odense University Hospital, 5000 Odense, Denmark; 8Department of Clinical Research, Faculty of Health Sciences, University of Southern Denmark, 5000 Odense, Denmark

**Keywords:** gestational diabetes mellitus, microRNAs, predictive biomarker, serum, circulating microRNA, miR-16-5p, miR-29a-3p, miR-134-5p, miR-122-5p, randomized control trial, obesity, pregnancy

## Abstract

Early identification of gestational diabetes mellitus (GDM) aims to reduce the risk of adverse maternal and perinatal outcomes. Currently, no circulating biomarker has proven clinically useful for accurate prediction of GDM. In this study, we tested if a panel of small non-coding circulating RNAs could improve early prediction of GDM. We performed a nested case-control study of participants from the European multicenter ‘Vitamin D and lifestyle intervention for GDM prevention (DALI)’ trial using serum samples from obese pregnant women (BMI ≥ 29 kg/m^2^) entailing 82 GDM cases (early- and late- GDM), and 41 age- and BMI-matched women with normal glucose tolerance (NGT) throughout pregnancy (controls). Anthropometric, clinical and biochemical characteristics were obtained at baseline (<20 weeks of gestation) and throughout gestation. Baseline serum microRNAs (miRNAs) were measured using quantitative real time PCR (qPCR). Elevated miR-16-5p, -29a-3p, and -134-5p levels were observed in women, who were NGT at baseline and later developed GDM, compared with controls who remained NGT. A combination of the three miRNAs could distinguish later GDM from NGT cases (AUC 0.717, *p* = 0.001, compared with fasting plasma glucose (AUC 0.687, *p* = 0.004)) as evaluated by area under the curves (AUCs) using Receiver Operator Characteristics (ROC) analysis. Elevated levels of individual miRNAs or a combination hereof were associated with higher odds ratios of GDM. Conclusively, circulating miRNAs early in pregnancy could serve as valuable predictive biomarkers of GDM.

## 1. Introduction

Gestational diabetes mellitus (GDM) is an important contributor to the unprecedented threat of Type 2 Diabetes (T2D) and obesity epidemic seen worldwide. Women with a previous diagnosis of GDM contribute substantially (up to 31%) toward the total female population with diabetes [1]. Moreover, exposure to intrauterine hyperglycemia confers a future increased risk of T2D and cardiometabolic diseases in offspring of GDM mothers [2,3,4].

Although GDM can occur at any time during pregnancy, GDM is most often diagnosed in the late second trimester or early in third trimester between the 24th–28th gestational weeks, which coincides with markedly increased insulin resistance [5,6,7,8]. Maternal, fetal and neonatal consequences of GDM include pre-eclampsia, pre-term birth, excessive fetal growth and neonatal hypoglycemia [9,10]. Indeed, excessive fetal growth in GDM predates the usual time of GDM diagnosis with fetuses of obese women showing abnormal growth already by 20 weeks of gestation [11]. Currently, no randomized controlled trials have investigated the beneficial effects diagnosing GDM prior to 24–28 weeks of gestation [12,13]. However, early identification of GDM, at least in high-risk individuals, and subsequent treatment could improve obstetric and perinatal outcome [14].

Since their discovery, microRNAs (miRNAs) have been a focus of translational clinical research also in regards to the pathophysiology of GDM [15]. Endogenous miRNAs have a high stability in body fluids [16], which is partly attributed to their enclosure in extracellular vesicles [17], and they are accessible with minimal non-invasive sampling [18]. Furthermore, miRNAs are ideal biomarker candidates as technologies used for detection are specific and sensitive, and may be multiplexed [19]. By post-transcriptional gene regulation, miRNAs modulate various physiological processes and their dysregulation is often implicated in disease [20,21]. The placenta expresses more than 500 miRNA species [22,23] attributable in part by two placenta expressed miRNA clusters; the chromosome 19 and 14 miRNA clusters; C19MC and C14MC, respectively. Dysregulated placenta-associated miRNAs [24,25,26,27] as well as altered circulating miRNA profiles [28,29,30,31,32,33,34,35,36,37,38,39,40,41,42,43] have been associated with GDM, mirroring physiological and pathological changes during pregnancy.

There is an urgent need for early identification of women developing GDM later in pregnancy, ideally using affordable and effective biomarkers, with the objective to initiate treatment earlier. Ultimately, using these biomarkers could facilitate a reduction of the harmful effects of hyperglycemia on the mother and fetus. The aim of the current study was to investigate the utility of eight selected miRNAs as early non-invasive predictive biomarkers of GDM. The intention is to predict GDM development independently of confounding risk markers such as obesity, advanced maternal age and offspring sex.

## 2. Materials and Methods

### 2.1. Participants

In the complete DALI lifestyle study (*n* = 639) [44], obese pregnant women with a pre-pregnancy body mass index (BMI) of ≥29 kg/m^2^, from across nine European countries were invited to participate if they were ≤19 ± 6 days of gestation (baseline), carried a singleton pregnancy, and were above 18 years. Women were excluded if they had pre-existing diabetes, GDM at inclusion according to IADPSG/WHO2013 criteria (oral glucose tolerance test (OGTT), venous plasma glucose: fasting ≥ 5.1 mmol/L, 1 h  ≥  10 mmol/L, 2 h  ≥  8.5 mmol/L), an inability to walk ≥100 m, chronic medical conditions or a psychiatric disorder. In women without GDM, the 75 g OGTT was repeated at 24–28 and 35–37 weeks and evaluated according to IADPSG/WHO2013 criteria [44].

Pregnant women for this nested case-control study were carefully age- and BMI-matched from the DALI lifestyle study. The lifestyle interventions were shown not to impact obstetric outcome nor the development of GDM [45,46]. Additionally, pregnant women, who were diagnosed with GDM at baseline and thus excluded from entering the DALI study, were included in the current study (thus only baseline data available). GDM cases were classified according to the time of GDM diagnosis; early GDM (before 20 weeks of gestation; baseline) and late GDM (at 24–28 weeks of gestation), while controls were normal glucose tolerant (NGT) throughout pregnancy. GDM diagnosis at 35–37 weeks was not used in the current study. A flowchart of the study if provided in Appendix A.

### 2.2. Sample Collection and Storage

Maternal clinical, biochemical and anthropometric assessment used in the current study have been previously described [44]. In brief, after a 10h overnight fast, blood samples were collected at baseline, prior to and during an OGTT. All samples were centrifuged and stored immediately at −20 °C or −80 °C before being processed in the ISO-certified central trial laboratory in Graz, Austria. Insulin concentrations were quantified using commercially available Enzyme-Linked Immuno Sorbent Assay (ELISAs) and insulin resistance was indirectly assessed using the homeostatic index of insulin resistance (HOMA-IR) index according to this formula: HOMA-IR = [Fasting plasma insulin (mU/L) **·** Fasting plasma glucose (mmol/L)]/22.5 [47]. Quantitative insulin sensitivity check index (QUICKI): 1/[log fasting plasma glucose (mg/dL) + log fasting plasma insulin (μU/mL)] [48]. Oral glucose insulin sensitivity (OGIS) index was derived from the OGTT using three formulas according to Mari et al. [49]. The Matsuda insulin sensitivity index [50] was calculated as follows: Matsuda = 10,000/square root of [(mean serum insulin **·** mean plasma glucose during OGTT) **·** (fasting plasma glucose **·** fasting serum insulin)]. Both HOMA-IR and QUICKI reflect the effects of insulin on hepatic glucose production (i.e., hepatic rather than peripheral insulin resistance) while the Matsuda index reflects whole body (hepatic and muscular) insulin sensitivity and the result obtained from the OGIS model reflects that of the hyperinsulinemic-euglyemic glucose clamp [51]. All other analytes were quantified by conventional chemistry methods.

### 2.3. Small Non-Coding RNA Isolation and Analysis

Eight small non-coding RNAs were pre-selected based on their reported association with GDM or features linked to a challenged metabolic profile (Appendix A). Blood samples collected at baseline (average gestational age 15.1 ± 2.4 weeks, *n* = 123) prior to the OGTT were used in this study.

Circulating total RNA was extracted from 250 µl serum using TRI^®^Reagent LS (Sigma-Aldrich, Søborg, Denmark) according to manufacturers’ protocol and with the addition of glycogen (15 nmol) as a carrier and synthetic ath-miR-159 (0.25 pmol) as a spike-in control. RNA quantity and purity was determined by NanoDrop ND-100 spectrophotometer (ThermoFisher Scientific, Hvidovre, Denmark) with quality control performed on selected samples using Agilent Small RNA chips on the Agilent 2100 Bioanalyzer (Agilent Technologies, Glostrup, Denmark).

A total of 100 ng serum RNA was reverse transcribed using miRNA specific stem-loop RT primers and MultiScribe™ Reverse Transcription kit (ThermoFisher Scientific) according to manufacturers’ protocol. Synthetic *Caenorhabditis elegans* (cel)-miR-39 was added (0.5 amol) as spike-in control for cDNA synthesis.

Diluted cDNA was mixed with appropriate primers and QuantiTect SYBR Green PCR master mix (Qiagen, Copenhagen, Denmark). All qPCR assays, in duplicates, were pipetted using an automated pipetting system consisting of two instruments; the PIRO^®^ (Dornier-LTF, Germany) and epMotion^®^96 (Eppendorf Nordic, Hørsholm, Denmark), respectively. The 384-well plates were analyzed using the ViiA real time PCR System (ThermoFisher Scientific) and calibrated using an interplate-calibrator consisting of pooled sample material. Raw CT-values were normalized against the geometric mean of three reference genes; the endogenous small nuclear U6, ath-miR-159 and cel-miR-39. miRNA quantities were determined using the standard curve method, and subsequently expressed relative to the levels of the NGT group.

### 2.4. Ethical Statement

Written informed consent was obtained at baseline. The study was performed according to the Declaration of Helsinki II and local ethics committees approved the study. Trial registration was ISRCTN70595832.

### 2.5. Statistical Analysis

Continuous variables were tested for normality of distribution, and log-transformed if skewed variables (BMI, miRNAs levels, variables pertaining to glucose metabolism as well as triglycerides and leptin levels) were used, where necessary, in subsequent analyses. Variables with approximately normal distribution are presented as mean ± standard deviation (SD), and those with skewed distribution are presented as median and interquartile range (IQR). Continuous variables of women with a normal glucose tolerance were compared with those of women, who had either GDM at baseline (early GDM) or who developed GDM later (late GDM), using independent t-test with Bonferroni correction for multiple testing. Categorical variables were evaluated using Χ^2^ test or Fishers’ exact test. Mean adjusted levels were obtained from multiple linear regression models and compared between the three different groups, after adjustment for the following covariates: maternal BMI and age at baseline, gestational age and offspring sex.

Binary logistic regression for the association between GDM and miRNA were carried out with adjustment for potentially confounding variables. Results are represented by the odds ratio (OR) and 95% confidence interval (CI). Receiver Operate Characteristic (ROC) curve analysis was performed and Area Under the Curve (AUC) was reported for individual and a combination of miRNAs.

Predicted target genes for the individual miRNAs were found using TargetScan (v.7.2, human, http://www.targetscan.org/vert_72) [52] and pathway enrichment analysis was performed using the PANTHER (Protein ANalysis THrough Evolutionary Relationships) Classification System (v.15.0, http://pantherdb.org) and the PANTHER pathways annotation set [53]. Enrichment analysis was visualized by RStudio (v. 1.3.1093, RStudio, PBC Boston, MA, USA, http://www.rstudio.com) using the ggplots2 (v.3.1.0, https://ggplot2.tidyverse.org) package [54].

A *p*-value of <0.05 was considered statistically significant. All analyses were conducted using SPSS, version 26.0 (SPSS Inc., Chicago, IL, USA) and graphical representation was done in GraphPad Prism vers. 9 (GraphPad Inc., La Jolla, CA, USA.). Graphical abstract was created with BioRender (Biorender.com).

## 3. Results

### 3.1. Anthropometric Baseline Measurement of Study Participants

Forty-one healthy pregnant, glucose tolerant women and 82 women with either an early (>20 weeks of gestation) or a late (at 24–28 weeks of gestation) diagnosis of GDM were studied. Importantly, all of the participants were well-matched in terms of age, BMI, gestational age, parity, smoking status, ethnicity and their lipid profiles (Table 1). Women with an early diagnosis of GDM (early GDM) had higher fasting glucose, fasting insulin, higher insulin resistance and lower insulin sensitivity at baseline compared with women with a later diagnosis of GDM (late GDM) (Table 1). Before 20 weeks of gestation, higher glucose levels were observed in women with late GDM compared to women with NGT. There were no differences in offspring birthweight or sex.

### 3.2. Non-Coding RNAs are Associated with GDM

Eight pre-selected circulating non-coding miRNAs were measured at the DALI study baseline. Women, who were diagnosed with late GDM presented with elevated levels of miR-16-5p (mean difference 1.1, *p*-adjusted 0.008), miR-29a-3p (mean difference 0.7, *p*-adjusted 0.004), and miR-134-5p (mean difference 0.5, *p*-adjusted 0.046) compared with women who remained NGT throughout gestation (Figure 1A–C). Additionally, two miRNAs were significantly different between the two GDM groups with women with late GDM displaying higher levels of miR-16-5p (mean 1.1 ± 1.1, *p*-adjusted 0.008) and miR-122-5p (mean 0.9 ± 1.1, *p*-adjusted 0.046), respectively (Figure 1A,D).

Four additional miRNAs (miR-103-3p, -223-3p, -330-3p and 433-3p) had circulating levels with a gradual increase found in GDM cases although not significant upon correction for multiple comparisons (Appendix A).

### 3.3. Correlations between miRNAs and Clinical and Biochemical Phenotypes

The relationship between circulating miRNAs and clinical variables measured at baseline as well as at 24–28 weeks and 35–37 weeks were investigated (Table 2). Among the selected miRNAs, miR-16-5p correlated positively with age (r = 0.203, *p* = 0.029, *n* = 116). Upon adjustment for confounding variables (gestational age, BMI and offspring sex) this correlation increased (r_adj_ = 0.221, *p*_adj_ = 0.028, *n* = 97). In response to a 2-h OGTT, intermediate 1-h glucose values measured at baseline or at 24–28 weeks correlated with miR-29a-3p (r = 0.191, *p* = 0.040, *n* = 116) and miR-16-5p (r = 0.256, *p* = 0.023, *n* = 79) respectively.

Three miRNAs (miR-29a-3p, -134-5p and -16-5p) correlated with 2-h fasting glucose levels measured at 24–28 weeks of gestation, and the degree of correlation increased after adjustment for maternal age and BMI, gestational age and offspring sex (miR-29a-3p, r_adj_= 0.325, *p*_adj_ = 0.004, miR-16-5p, r_adj_ = 0.368, *p*_adj_ = 0.001, miR-134-5p, r_adj_ = 0.241, *p*_adj_ = 0.033). Glycated hemoglobin (HbA1c) measured at 24–28 weeks correlated with baseline miR-16-5p (r = 0.259, *p* = 0.031, *n* = 69). A negative association between insulin sensitivity through the OGIS index and miR-122-5p was observed (r = −0.437, *p* = 0.004, r_adj_ = −0.499, *p*_adj_ = 0.002). Other measurements of insulin sensitivity such as the Matsuda index also showed negative correlation with miR-122-5p. Of note, HOMA-IR was not correlated with the selected miRNAs. Both HDL cholesterol (r = −0.240, *p* = 0.043, *n* = 72) and leptin (r = −0.236, *p* = 0.036, *n* = 79) levels were negatively correlated with miR-122-5p. Birthweight was positively associated with miR-122-5p (r = 0.243, *p* = 0.013, r_adj_ = 0.237, *p*_adj_ = 0.018). All of the miRNAs correlated to various degrees with each other.

### 3.4. Classification of GDM Cases Using Serum miRNAs

To evaluate the performance of serum miRNAs as predictive biomarkers for GDM, i.e., comparing women with NGT with late GDM, ROC curves were constructed using the miRNAs, which were different between the two groups (Figure 2A–C). When assessing individual miRNAs, the discriminatory capability of miR-29a-3p and miR-16-5p had the highest area under the curves (AUCs) of 0.698 (sensitivity of 72.5% and specificity of 57.5%, *p* = 0.002) and 0.687 (sensitivity of 60.5% and specificity of 57.5%, *p* = 0.005), respectively. This was followed by miR-134-5p with an AUC of 0.654 (sensitivity of 65.0% and a specificity of 51.2%, *p* = 0.017).

Using binary logistic regression, three (miR-16-5p, -29a-3p and -134-5p) miRNAs were incorporated into a 3-miRNA-signature increasing the AUC to 0.717 (sensitivity of 63.2% and specificity of 66.7%, *p* = 0.001) (Figure 2D). In comparison, clinically accepted glucose values such as baseline FPG or 2h glucose values as predictors of GDM had AUCs of 0.687 (sensitivity of 65.0% and specificity of 61.0%, 95% CI 0.56–0.80) and 0.681 (sensitivity of 65.9% and specificity of 56.1%, 95% CI 0.57–0.80), respectively. The 1h glucose and HbA1c values were not able to distinguish late GDM cases from NGT (Appendix A) at baseline at least. If the 3-miRNA-signature was combined with FPG an additional increase in the AUC to 0.81 was observed (sensitivity of 68.4% and specificity of 82.1%, 95% CI 0.71–0.90) (Figure 2D). Increased discriminatory power was also observed when the 3-miRNA signature was combined with 2h glucose values (AUC = 0.764, 95% CI 0.65–0.88, data not shown).

We also investigated if risk markers previously identified in the DALI study would improve the predictive power of our 3-miRNA signature. If maternal resting heart rate (RHR), neck circumference or maternal height were added individually to the 3-miRNA signature equal increases in the AUC was seen (AUC 0.727, 95% CI 0.61–0.84, data not shown).

The effects of the individual miRNAs or a combination hereof on the likelihood of being diagnosed with GDM were investigated using logistic regression. With a one-unit increase in miR-29a-3p, the odds ratio of being diagnosed with GDM was 2.5 (95% CI 1.3–4.5, *p* = 0.003), while both miR-16-5p and miR-134-5p was associated with odds ratios of 1.9 (95% CI 1.2–3.1, *p* = 0.006) and 1.9 (95% CI 1.1–3.5, *p* = 0.027) respectively (Figure 3). At baseline, miR-122-5p levels did not affect the odds ratio of a GDM diagnosis (OR 1.2, 95% CI 0.9–1.6, *p* = 0.17). The combined 3-miRNA signature had an odds ratio of a GDM diagnosis of 85.5 (95% CI 5.89–1248.1, *p* = 0.001) with larger confidence intervals resembling the range of plausible odds ratios, when the three miRNAs are combined. Importantly, addressing potential confounding effects of maternal age and BMI as well as offspring sex did not alter the increased odds ratio of GDM with either miRNA (Figure 3).

Analyzing whether individual miRNAs could be used for diagnosing GDM cases at baseline resulted in modest AUCs (Appendix A). However, two miRNAs, miR-16-5p and miR-122-5p were able to distinguish between women, who were diagnosed with GDM at baseline compared to those diagnosed at a later time in pregnancy (Appendix A).

### 3.5. Predicted Targets and Pathway Analysis

Key to understanding the potential role of the identified miRNAs in relation to GDM is to investigate, which predicted targets and pathways are affected. A total of 1890 unique target transcripts were identified with the majority of them linked to miR-29a-3p targets. No overlap between the in silico predicted targets and all four miRNAs were observed. However, 148 targets were common between two or more miRNAs (Figure 4A).

Pathways associated with vascular endothelial growth factor (VEGF)-, fibroblast growth factor (FGF)-, phosphoinositide (PI)-3 kinase-, Notch- and insulin signaling were all positively enriched by more than 4.5-fold (Figure 4B). The biosynthesis of S-adenosylmethionine was the most enriched pathway by 45-fold (*p* = 0.028) due to the methionine adenosyltransferase 2A (MAT2A) transcript which is a predicted target of both miR-29a-3p and 134-5p. Predicted target genes in the enriched pathways are all listed in Appendix A.

## 4. Discussion

Currently, there are no good predictive biomarkers for the development of GDM. In this study, overweight and obese women who were NGT at baseline but diagnosed with GDM at 24–28 weeks of gestation had elevated miR-16-5p, -29a-3p and -134-5p levels already prior to 20 week of gestation. Furthermore, a combination of the three miRNAs into a 3-miRNA signature (miR-16-5p, -29a-3p and -134-5p) showed high predictive discriminatory power comparable with clinical accepted glucose values (FPG, 1-h and 2-h glucose values after an OGTT). Additionally, miRNA expressions were associated with increased odds ratios of GDM.

Consistent with our results, elevated circulating miR-16-5p was also shown by Zhu et al. [55] using peripheral blood collected from Chinese women early in pregnancy (16–19 weeks of pregnancy), although ROC was not constructed. Additionally, fetal macrosomia was also associated with elevated plasma levels of miR-16-5p collected between 18–28 weeks of gestation [56]. In our nested case-control study, both miR-16-5p and miR-29a-3p was also associated with macrosomia (Appendix A).

In support of our study, the temporal profile of plasma miR-16-5p measured at 16–20 weeks, 20–24 weeks and 24–28 weeks’ of gestation in Chinese women gradually increased throughout pregnancy [57]. Absolute values of plasma miR-16-5p at 24–28 weeks of gestation clearly distinguished between GDM cases and controls (with cut-off >2554 copies, sensitivity 41.6% and specificity 95.8%) [57]. However, Cao et al. did not investigate the utility of miR-16-5p for early identification of GDM in their study [57]. Two other studies investigated the expression of miR-16-5p in third-trimester samples of maternal leukocytes from Turkish pregnant women with pre-eclampsia and GDM [32] or from pregnant women with polycystic ovary syndrome and GDM [31]. They showed no differences in the expression of miR-16-5p between the different groups. Of note, the GDM and control group in the two Turkish studies [31,32] were the same. Similarly, no differences in miR-16-5p levels were observed from second-third trimester serum samples from Mexican women with GDM [34] or serum sampled over a large part of gestation (13–31 weeks, median; 27 weeks) from South African women with GDM [42]. Discrepancies between studies could be due to sampling period, sampling type and ethnicity. It is also worth considering advanced maternal age as a confounder as this is associated with an increased risk of GDM [9]. It has been suggested that miRNA levels might also change in an age-dependent fashion. Indeed, in this study maternal age positively correlated with miR-16-5p in this study. Despite this, miR-16-5p was still associated with increased odds ratios of GDM after adjusting for maternal age. Of note, miR-16 is highly abundant in erythrocytes, and an upregulation of miR-16 has been associated with hemolysis [58]. However, no visual signs of hemolysis were observed in our samples.

While circulating miR-29a is increased in type 2 diabetes patients [59] few studies have investigated miR-29a in relation to GDM [33,34,37]. Consistent with our study, serum levels of miR-29a-3p obtained at 18–23 weeks of gestation was found to be higher in Mexican women with GDM according to the IADPSG criteria [34]. In white non-Hispanic women with a male offspring, circulating miR-29a-3p obtained at 7–23 weeks of gestation was associated with an increased risk of GDM [38]. We did not observe a sex-dependent expression of miR-29a-3p (Appendix A). By contrast, miR-29a was found decreased in serum collected from Chinese women at 16–19 gestational weeks compared to controls [33]. Moreover, since increased levels of miR-29a have also been related with both preeclampsia and gestational hypertension [28,60], it would be relevant to investigate whether miR-29a could be of use as a general prognostic factor for complicated pregnancies.

The exact cellular sources of the circulating miR-29a-3p, studied here as another early pregnancy GDM risk marker, are not known. In the current study, we did not evaluate to which degree the content of circulating exosomes contributed to the individual miRNA levels. However, extracellular vesicle miR-29a content is increased in women with GDM [28]. Additionally, a greater increase in circulating placenta-derived exosomes across gestation was observed in GDM pregnancies compared to normal pregnancies [61]. Further, consistent patterns of selected miRNAs were demonstrated across placenta-derived exosomes, plasma exosomes and placenta exosomes isolated from GDM pregnancies [41].

Liver [62], skeletal muscle [63], beta cell [64] and adipose tissue [65] miR-29a levels are increased in states of insulin resistance and hyperglycemia. Accordingly, circulating miR-29a decreases upon metformin treatment [66]. Therefore, it is conceivable that the increased early pregnancy miR-29a levels, as well as the other differentially expressed miRNAs in women who later develop GDM, could be an indicator of a general impaired metabolic state. MiR-29a along with miR-134 is predicted to target Methionine Adenosyltransferase 2A (MAT2A). Inhibition of MAT2A by miR-29a and miR-134, and deficiency of S-adenosylmethionine (SAM) in pregnancy could have implications for fetal programming, because SAM constitutes the required methyl donor in epigenetics [67].

Predicted mRNA targets of the four miRNAs showed significant enrichment of pathways related to feto-placental development, such as VEGF signaling and angiogenesis [68], the insulin/insulin-like growth factor (IGF) signaling pathway [69], as well as development in general (p38MAPK pathway, FGF signaling, PI3 kinase pathway, Ras pathway and Notch signaling). Moreover, one-carbon metabolic pathways related to fetal programming and epigenetics such as SAM biosynthesis and *de novo* pyrimidine ribonucleotides biosynthesis [70] are also predicted targets by the altered miRNAs. In addition, pathways related to the innate (phenylethylamine degradation) and adaptive immune system (T cell activation) were also predicted targets of the GDM-associated circulating miRNAs, suggesting possible moderation of placental immune tolerance by these miRNAs [71]. Thus, important pathways related to fetal development, epigenetic programming and the immune system could be altered by the miRNAs increased in the maternal circulation in GDM.

Comprehensive profiling of maternal and fetal circulating miRNAs demonstrated elevation in pregnant women and cord blood of miR-134-5p. It belongs to the chromosome 14 miRNA cluster and is highly expressed in placenta tissue [72]. After birth, localization of miR-134-5p is restricted to the brain [73]. In line with our finding, hyperglycemia induces miR-134-5p expression in endothelial progenitor cells and overexpression of miR-134-5p results in impaired tube formation and cell migration [74]. In our study, we observed elevated levels of miR-134-5p at baseline in those women, who were diagnosed with GDM later in pregnancy. Interestingly, in primary feto-placental endothelial cells (fpEC) isolated from obese women with GDM, miR-134-5p showed a sex-dependent regulation in women carrying female offspring [27]. However, we were not able to confirm a sex-dependent expression of our selected circulating miRNAs (Appendix A). Fetal sex may influence the risk of GDM [75]. Hence, we adjusted for offspring sex. However, in clinical practice this covariate would often not be used despite the fact that the sex of the fetus can be known with high certainty at the mid-pregnancy ultrasound scans.

Additionally, miR-134-5p has also been associated with the progression of pre-eclampsia [76] and with the inhibition of trophoblast cell infiltration [77]. Tumor necrosis factor alpha (TNF-α), a cytokine produced in the placenta and adipose tissue, is elevated in late pregnancy and associated with insulin resistance [78]. Induction of miR-134-5p, by TNF-α, has been observed [79] and suggests a potential role for miR-134-5p in insulin-mediated glucose disposal and insulin sensitivity.

The women included in this study were all overweight/obese with GDM diagnosed around 24–28 gestational weeks showing elevated levels of the liver-enriched miR-122-5p already at baseline. This result is in line with the study by Gillet et al. which showed that pregnancies affected by GDM were associated with elevated serum exosomal miR-122 early in gestation (6–15 weeks of gestation) [28]. Several studies have shown that increased miR-122 is associated with obesity, insulin resistance, metabolic syndrome [80,81,82], while metformin treatment [66] and weight loss [83] results in a reduction of miR-122-5p. We demonstrate in our study that insulin sensitivity, measured through OGIS, and HDL cholesterol were negatively correlated with miR-122-5p, while offspring birthweight was positively correlated with miR-122-5p, thus further strengthening the role of miR-122 as a marker of impaired metabolic health.

In contradiction to our study, Carreras-Badosa et al. found that maternal pre-gestational obesity and gestational weight gain were associated with decreased levels of second trimester (24–32 weeks of gestation) plasma miR-122-5p [36]. Moreover, a recent study on intrauterine fetal programming and GDM showed decreased circulating maternal and fetal miR-122-5p in a GDM rat model [84]. The reason for these discrepancies is not known, but could be due to differences in ethnicity of patients and interspecies differences in miRNA regulation.

The current IADPSG/WHO criteria have not been validated for use early in pregnancy. In the full DALI study, the majority of women, diagnosed with GDM before 20 weeks of gestation were identified by increased fasting glucose and found to exhibit features of the metabolic syndrome compared with women with normal glucose tolerance at baseline [85]. Despite an observed mild hyperglycemic state at baseline in the early GDM group retrospectively, it was speculated whether the diagnostic threshold for the full DALI study should have been higher thus including more of the early GDMs in the study [85]. It is recognized that considerable heterogeneity exists between women with GDM [86,87,88]. Phenotypical intergroup differences could, therefore, partly explain why most of the miRNAs in our study in the early GDM and NGT groups were similar and distinct from those women developing GDM de novo later in pregnancy.

Other potential biomarker candidates for early diagnosis of GDM, for prediction of GDM development >24 weeks of gestation and independent risk factors of GDM have been evaluated in the DALI study. These include plasma-glycated CD59 (pGCD59) [89], glycated hemoglobin (HbA1c) [90] and non-modifiable, clinical characteristics such as maternal resting heart rate (RHR), neck circumference and height [91]. Plasma-GCD59 obtained before 20 weeks of gestation showed moderate prognostic value for identifying women who would develop GDM later in pregnancy (AUC = 0.68, 95% CI, 0.64–0.73, *n* = 77) [89]. Moderate predictive value was partly ascribed to the need of some degree of hyperglycemia for glycation of pGCD59 to happen [89]. Post hoc analysis of HbA1c within the entire DALI study (AUC 0.55) [90], and in our nested study (AUC 0.60, *p* = 0.16) showed poor sensitivity for detecting GDM at any time in pregnancy. However, combing FPG with the 3-miRNA signature resulted in a higher AUC, as well as higher sensitivity and specificity. This is in agreement with a recent study by Hoorn et al. [92] who concluded that the best performing prognostic model for early prediction of GDM includes both clinical characteristics and early glucose measurements.

A limitation of the current study is that only selected miRNAs were investigated and other potential GDM prediction biomarkers could have been missed. Another caveat is that the DALI study only included high-risk overweight/obese women of largely European descent. Although obesity is a major risk factor of GDM, other factors such as insulin secretory defects contribute to the development of GDM which is evident among women with normal BMI. Despite these limitations, the results of this study suggest that a combination of circulating miRNAs measured before 20 weeks of gestation can predict incident GDM diagnosed between 24–28 weeks of gestation.

The findings in the present study needs to be validated in larger independent cohorts. Preferentially, the predictive utility of the 3-miRNA signature combined with early glucose measurements should be evaluated in both lean and obese women. For miRNAs to be used routinely in the clinic, they need to perform better than the current accepted glucose values, as well as be cost-effective with high-throughput. Additionally, they should not be influenced by the fasting-state of the woman.

## 5. Conclusions

This study provides evidence to justify future evaluation of circulating miRNAs early in pregnancy and their potential as screening/predictive biomarkers. The importance of identifying biomarkers that are not influenced by confounding variables and type of laboratory method capable of identifying women at risk of GDM early in pregnancy is clearly valuable and would greatly improve preventive strategies. Here, we identify three miRNAs (miR-16-5p, -29a-3p and miR-134-5p) in high-risk individuals as being elevated in early pregnancy in women who develop GDM. These miRNAs both individually and in combination are associated with higher odds of GDM. Early identification of GDM would most likely improve short-term pregnancy outcomes. It would be of interest to investigate if these miRNAs are already increased before pregnancy.

## Figures and Tables

**Figure 1 cells-10-00170-f001:**
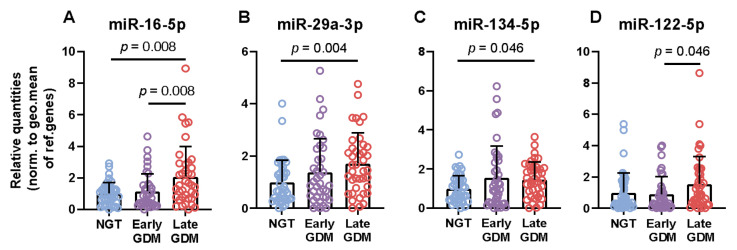
Circulating miRNAs are elevated in late GDM cases. Relative mean quantities with standard deviations of serum microRNAs expressed in the three groups (**A**–**D**). Data were normalized against the geometric mean of snRNA U6, ath-miR-159 and c.el.miR-39. All indicated *p*-values were determined by independent t-test on logarithmic transformed data with Bonferroni correction for multiple comparisons.

**Figure 2 cells-10-00170-f002:**
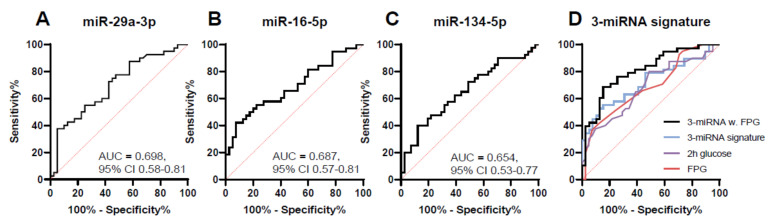
Predictive capabilities of circulating miRNAs. ROC curves based on miRNA levels. Shown are the three miRNAs distinguishing between GDM cases diagnosed later in pregnancy and NGT cases (**A**–**C**) as well as a combination of these miRNAs ((**D**) 3-miRNA signature, miR-16-5p, -29a-3p and -134-5p, AUC = 0.717, 95% CI 0.60–0.83). Fasting plasma glucose (FPG, AUC = 0.687, 95% CI 0.57–0.80) and 2-h glucose value (2h glucose, AUC = 0.681, 95% CI 0.57–0.80) were evaluated alone or in combination with the 3-miRNA signature.

**Figure 3 cells-10-00170-f003:**
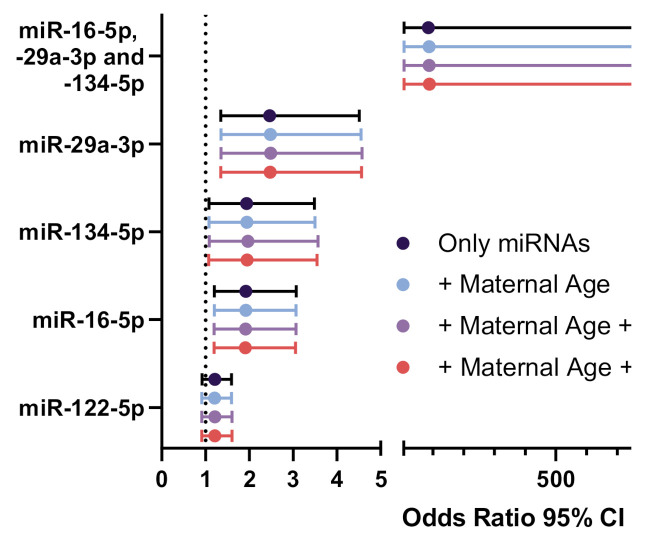
Odds ratios for the association of miRNAs with GDM. Odds ratio plot showing the predictive effects of either a single miRNA, a combination of these, or miRNAs after adjustment for confounding variables such as maternal age and BMI and offspring sex on GDM diagnosis. Normal glucose tolerant women were compared to women who were diagnosed with GDM later in pregnancy (*N* = 77–81). BMI: Body mass index.

**Figure 4 cells-10-00170-f004:**
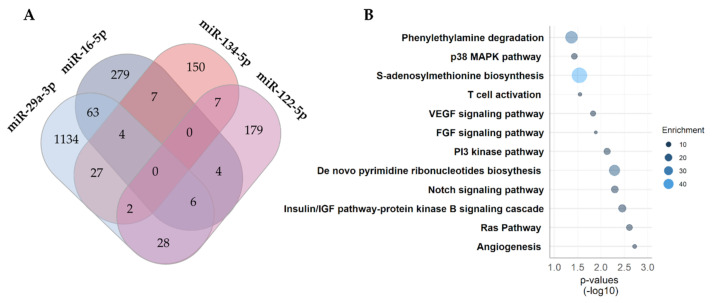
Predicted targets and pathway enrichment analysis. Predicted targets of the four differentially expressed miRNAs associated with GDM. (**A**) Venn diagram of predicted targets identified through TargetScan v.7.2. (human). (**B**) Enrichment analysis of predicted targets common to two or more miRNAs. The size and the color of the circle corresponds to the fold enrichment.

**Table 1 cells-10-00170-t001:** Baseline characteristics for anthropometric and clinical parameters of study participants measured prior to 20 gestational weeks.

	NGT (*n* = 41)	Early GDM (*n* = 41)	Late GDM (*n* = 41)	NGT vs. Early GDM	NGT vs. Late GDM	Early GDM vs. Late GDM
Gestational age at inclusion (weeks)	15.2 ± 2.4	14.9 ± 2.4	15.3 ± 2.5	1.0	1.0	1.0
Gestational age at delivery (weeks)	40.1 ± 1.1	39.3 ± 1.7	40 ± 39.3	0.17	0.67	0.92
Age (years)	33.2 ± 3.8	33.7 ± 4	32.7 ± 4	1.0	1.0	0.73
BMI at inclusion (kg/m^2^)	33.3 (32.2–35.4)	33.3 (31.7−36.0)	33.3 (31.7−35.9)	0.99	0.99	0.99
Waist circumference (cm)	107 ± 10.7	107.7 ± 9.8	105.6 ± 8.8	1.0	1.0	1.0
Neck circumference (cm)	36.0 (35.2–37.0)	36.0 (34.5−38.0)	35.5 (34.6−36.9)	0.58	0.58	0.58
Resting heart rate (RHR)	79.7 ± 11.3	83.9 ± 12.2	80.0 ± 9.3	0.28	1.0	0.32
Fasting glucose (mmol/L)	4.7 (4.3–4.8)	5.2 (5−5.6)	4.9 (4.6−5.2)	**<0.001**	**0.024**	**0.002**
1h glucose (mmol/L)	7.2 (5.9–7.6)	9.0 (7.2−10.3)	7.2 (6.2−8.3)	**<0.001**	1.0	**0.002**
2h glucose (mmol/L)	5.6 (4.9–6.5)	7.3 (6.2 − 8.4)	6.4 (5.7−7.4)	**<0.001**	**0.024**	0.14
HbA1c (%)	5.0 (4.8–5.4)	5.1(5−5.4)	5.2 (5.0−5.5)	0.16	0.16	0.16
Fasting insulin (μU/mL)	11.9 (9.7–18.4)	18.1 (13.9−25.8)	14.6 (9.4−18.5)	**0.004**	1.0	**0.027**
1h insulin (μU/mL)	95.2 (57.5–161.1)	127.7 (66.2−173)	95.8 (52.5−175.5)	0.45	0.45	0.45
2h insulin (μU/mL)	52.2 (34.5–88.1)	92.4 (58.4−153.7)	59.8 (46.9−100.7)	**0.001**	0.59	0.078
HOMA-IR	2.61 (1.98−3.70)	4.25 (3.19−6.53)	3.11 (2.13−4.19)	**<0.001**	0.87	**0.006**
Quicki	0.330 ± 0.02	0.309 ± 0.021	0.325 ± 0.027	**<0.001**	0.94	**0.005**
OGIS (ml/min/m^2^)	424.9 ± 56.9	328.7 ± 67.7	385.1 ± 52.8	**<0.001**	0.10	**0.022**
Matsuda	2.98 (2.3−4.4)	2.1 (1.0−3.0)	2.83 (2.1−3.9)	0.061	1.0	0.61
Triglycerides (mmol/L)	1.32 (1.13−1.60)	1.48 (1.07−2.04)	1.27 (0.9−1.68)	0.32	0.32	0.32
Free fatty acids (mmol/L)	0.61 ± 0.19	0.62 ± 0.22	0.65 ± 0.17	1.0	1.0	1.0
HDL cholesterol (mmol/L)	1.46 ± 0.23	1.4 ± 0.27	1.49 ± 0.26	0.93	1.0	0.34
LDL cholesterol (mmol/L)	3.16 ± 0.81	2.98 ± 0.77	3.32 ± 0.75	0.94	1.0	0.17
Leptin (pg/mL)	36.0 (25.2−46.1)	31.9 (23.1−43.6)	38.9 (29.5−47.3)	0.53	0.53	0.53
Ethnicity (European/non-European)	37/4	33/8	32/9	0.35	0.23	1.0
Parity (Primipara/Multipara)	18/23	24/17	26/15	0.27	0.12	0.82
Previous GDM (No/Yes)	32/1	22/5	21/0	0.081	1.0	0.059
Smoking (No/Yes)	37/4	36/5	36/5	1.0	1.0	0.92
Offspring sex (Male/Female)	23/18	11/15	20/21	0.54	0.66	0.63
Birthweight (g) *	3548 ± 478	3441 ± 598	3557 ± 496	1.0	1.0	1.0

Data are presented as mean ± standard deviations or median (interquartile range). NGT, Normal glucose tolerance. GDM gestational diabetes. Continuous data were analyzed by either one-way ANOVA followed by Bonferroni correction if data were normally distributed or Kruskal-Wallis test or Mann-Whitney test for non-normally distributed data. Categorical data were analyzed by Chi-Square test (2-sided) or Fisher’s exact test. * *N* = 26 for early GDM offspring as these women were excluded from the DALI trial and follow-up was not complete. Early GDM only had values for the OGTT at baseline, as these women were excluded in the complete DALI study. HOMA-IR: Homeostatic index of insulin resistance, QUICKI: Quantitative insulin sensitivity check index, OGIS: Oral glucose insulin sensitivity index, Matsuda: Matsuda insulin sensitivity index. Bold font indicates statistical significance.

**Table 2 cells-10-00170-t002:** Significant Pearsons correlations between clinical variables and circulating miRNAs.

	miR-29a-3p	miR-134-5p	miR-16-5p	miR-122-5p
Age ^a^	0.133	0.114	**0.203 ***	0.077
1h glucose ^a^	**0.191 ***	0.101	**0.256 *^,b^**	0.136
2h glucose ^b^	**0.266** *	**0.226***	**0.345 ****	0.069
HbA1c ^b^	0.211	0.044	**0.259 ***	0.016
Matsuda index ^b^	0.125	0.131	0.082	**−0.309 ***
Insulin sensitivity (OGIS) ^b^	−0.040	0.016	−0.054	**−0.437 ****
Triglycerides ^a^	−0.088	−0.106	**−0.197 ***	0.030
Free fatty acids ^a^	**0.186 ***	0.059	−0.006	0.034
HDL cholesterol ^c^	−0.085	−0.091	−0.106	**−0.240 ***
Leptin ^c^	0.102	0.132	0.058	**−0.236 ***
Birthweight	0.061	0.068	0.047	**0.243 ***
miR-134-5p ^a^	**0.731 *****	-	-	-
miR-16-5p ^a^	**0.487 *****	0.181	-	**-**
miR-122-5p ^a^	0.173	0.050	**0.343 *****	-

Table 2 Association between clinical variables and circulating miRNAs. Clinical variables were measured at baseline ^a^, at 24–28 week ^b^, or at 35–37 week ^c^ and correlated with baseline miRNA levels. Pearson correlation coefficients are listed. Statistically significant values are in bold with *p* values, * *p* < 0.05, ** *p* < 0.01 and *** *p* < 0.001. Levels of miRNAs were normalized against the geometric mean of snRNA U6, ath-miR-159 and c.el.miR-39 and logarithmic transformed prior to analysis. Covariate were maternal age and BMI, gestational age and offspring sex.

## Data Availability

The raw data supporting the conclusions of this manuscript will be made available by the authors, without undue reservation, on request to the corresponding author.

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
