# Peer review of "The Predictive Value of miR-16, -29a and -134 for Early Identification of Gestational Diabetes: A Nested Analysis of the DALI Cohort"

_cells, 2021, doi:10.3390/cells10010170_

Round 1

Reviewer 1 Report

The MS submitted by Anja Elaine Sørensen and colleagues aims to investigate the prognostic value of several miRNA for early identification of gestational diabetes.

  1. I recommend the author to add a flowchart.
  2. The early and late stages described in the article are a bit confusing. Usually, the disease is diagnosed in 24-28 weeks. According to what criteria were Early GDM diagnosed in this study?
  3. Please list the time when blood samples were collected in this study, i.e., the gestational age of the participants.
  4. In Figure 1, in the Early GDM group, compared with NGT, these miRNAs did not increase. How to explain?
  5. In the Results section, the author carried out the enrichment analysis of the predicted targets and pathways. Therefore, they should briefly discuss them in the Discussion section.

Author Response

We thank the reviewers for their time contribution to improving our manuscript. We are very grateful for these constructive comments and thoughtful suggestions. We have done our best to accommodate the changes pointed out by the reviewers. Below is our point-by-point response to the comments raised by the reviewers with page and line-numbers for the associated changes introduced in the manuscript. Changes are made with track-change settings for easy identification.

Reviewer 1

  • I recommend the author to add a flowchart.

We thank the reviewer for the great suggestion. A flowchart has been added as a Supplementary Figure 1. The subsequent supplementary figure numbers have been changed accordingly.

  • The early and late stages described in the article are a bit confusing. Usually, the disease is diagnosed in 24-28 weeks. According to what criteria were Early GDM diagnosed in this study?

This is an important point. A diagnosis of GDM at baseline (Early GDM) or at 24-28 weeks of gestation (Late GDM) was made according to IADPSG/WHO2013 criteria (OGTT, venous plasma glucose: fasting ≥ 5.1 mmol/L, 1 h ≥ 10 mmol/L, 2 h ≥ 8.5 mmol/L). Please see page 2, line 88-92 and line 97-98. With the addition of the flow chart, Supplementary Figure 1, we hope the classification of the two groups should be clearer. 

  • Please list the time when blood samples were collected in this study, i.e., the gestational age of the participants.

The isolation of miRNAs was made with blood samples collected at baseline (average gestational age 15.1±2.4 weeks). This has been revised on page 3, line 117-118.   

  • In Figure 1, in the Early GDM group, compared with NGT, these miRNAs did not increase. How to explain?

We cannot explain why the levels of the individual miRNAs between NGT and Early GDM are similar. However, it is recognized that heterogeneity between GDM cases exists. In order to emphasize these phenotypical intergroup differences, we have added this point and three references to support this (ref. 85-87). We speculate that the Early GDM group is of a different metabolic phenotype than the late GDM group and that our miRNA signature is specifically able to identify women with the late GDM phenotype.

  • In the Results section, the author carried out the enrichment analysis of the predicted targets and pathways. Therefore, they should briefly discuss them in the Discussion section.

We agree with the reviewer. Hence, we have elaborated the discussion of selected pathways.

Reviewer 2 Report

Sørensen et al. performed a nested case-control study evaluating the possible role of several microRNAs as predictive biomarkers for GDM. Elevated baseline (<20 weeks of gestation) serum levels of three of them, namely miR‐16‐5p, ‐29a‐3p, and ‐134‐5p, significantly predicted the development of GDM, suggesting that these microRNAs might be valuable candidate biomarkers for GDM at a very early stage of pregnancy. Furthermore, the authors observed significant correlations between circulating microRNA levels and some clinical and biochemical parameters. The study is methodologically well conducted, the performed analyses are elaborated and the topic is overall interesting, as there is increasing interest in finding new markers helping understand the pathophysiology of GDM and with potential diagnostic, prognostic and therapeutic utility.

There are some minor issues to be addressed:

  • Line 40: I suggest using “predictive biomarkers” throughout the text instead of “prognostic biomarkers”, because the term “prognostic” generally refers to GDM outcomes, whereas these molecules were evaluated as candidate diagnostic biomarkers for GDM
  • Line 54: I suggest adding relevant references about the outcomes of pregnancy complicated by GDM (such as J Endocrinol Invest. 2018 Jun;41(6):671-676. doi: 10.1007/s40618-017-0791-y)
  • Line 66: There is growing interest in the possible role of microRNAs in the pathophysiology of GDM and GDM-related complications. I suggest adding a short sentence with a relative reference to underline the importance of evaluating microRNA expression in GDM compared to physiological pregnancy (Int J Mol Sci. 2020 Jun 4;21(11):4020. doi: 10.3390/ijms21114020) and about the role of gestational tissues and adipose tissue in the development of insulin resistance in pregnancy (such as Acta Diabetol. 2019 Jun;56(6):681-689. doi: 10.1007/s00592-019-01304-x)
  • Lines 80-87: Did the study involve women who had experienced GDM in a previous pregnancy? If yes, please specify the prevalence in each group
  • Line 173: “glucose level” should be replaced with “glucose levels”
  • Table 1: please add all the p-values, even those referring to not significant differences
  • Line 177: “presented at” should be replaced with “presented as”
  • Line 179: “normal disturbed” should be replaced with “normally distributed”
  • Line 180: “non‐normal distributed” should be replaced with “non-normally distributed”
  • Line 206: please specify the confounding variables for which the correlation has been adjusted
  • Line 291: please explain the acronyms and correct the acronym “VEFG‐”. Check the other acronyms throughout the text
  • Line 326: “Despite of this” should be replaced with “Despite this” or “in spite of this”
  • Line 343: microRNAs in circulating exosomes have not been evaluated in this study. Please clarify that exosomes are another source of microRNAs, which is quite different from evaluating them in serum
  • Line 373: a possible limit of this study is the fact that only obese women were included. Although obesity is a major risk factor for GDM, there are several other risk factors that clearly contribute to the development of this condition, which, in fact, develops even in women with normal BMI. Further studies should evaluate the possible role of these microRNAs as predictors of GDM even women with normal BMI

Author Response

We thank the reviewers for their time contribution to improving our manuscript. We are very grateful for these constructive comments and thoughtful suggestions. We have done our best to accommodate the changes pointed out by the reviewers. Below is our point-by-point response to the comments raised by the reviewers with page and line-numbers for the associated changes introduced in the manuscript. Changes are made with track-change settings for easy identification.

Reviewer 2

  • Line 40: I suggest using “predictive biomarkers” throughout the text instead of “prognostic biomarkers”, because the term “prognostic” generally refers to GDM outcomes, whereas these molecules were evaluated as candidate diagnostic biomarkers for GDM

This is a very good suggestion from the reviewer. We have changed the term “prognostic” to “predictive” throughout the manuscript where it was found appropriate.

  • Line 54: I suggest adding relevant references about the outcomes of pregnancy complicated by GDM (such as J Endocrinol Invest. 2018 Jun;41(6):671-676. doi: 10.1007/s40618-017-0791-y)

This reference has been added (ref 10).

  • Line 66: There is growing interest in the possible role of microRNAs in the pathophysiology of GDM and GDM-related complications. I suggest adding a short sentence with a relative reference to underline the importance of evaluating microRNA expression in GDM compared to physiological pregnancy (Int J Mol Sci. 2020 Jun 4;21(11):4020. doi: 10.3390/ijms21114020) and about the role of gestational tissues and adipose tissue in the development of insulin resistance in pregnancy (such as Acta Diabetol. 2019 Jun;56(6):681-689. doi: 10.1007/s00592-019-01304-x)

The review suggested by the reviewer has been added, page 2 (ref. 15) with supportive text.

While we find that the second paper suggested by the reviewer is important for the further understanding of GDM, we have carefully considered but chosen not to include this as a reference, because the current study focuses on circulating miRNAs as potential biomarkers for GDM. We acknowledge that the different gestational tissues could be important contributors to the total miRNA pool. However, evaluation of this was beyond the scope of the current study.   

  • Lines 80-87: Did the study involve women who had experienced GDM in a previous pregnancy? If yes, please specify the prevalence in each group

The number of women diagnosed with GDM in a previous pregnancy has been added to the baseline characteristic Table 1 with appropriate statistics. We observe no difference between the prevalence of prior GDM.

  • Line 173: “glucose level” should be replaced with “glucose levels”

This has been corrected, line 175.

  • Table 1: please add all the p-values, even those referring to not significant differences

All of the p-values have been added to Table 1. The significant results have been emphasized by making these bold.  

  • Line 177: “presented at” should be replaced with “presented as”

This has been corrected.

  • Line 179: “normal disturbed” should be replaced with “normally distributed”

This has been corrected.

  • Line 180: “nonnormal distributed” should be replaced with “non-normally distributed”

This has been corrected.

  • Line 206: please specify the confounding variables for which the correlation has been adjusted

We have added in parentheses the three confounding variables (maternal BMI, gestational age and offspring sex) for which the correlations have been adjusted for, page 6.

  • Line 291: please explain the acronyms and correct the acronym “VEFG”. Check the other acronyms throughout the text.

We are grateful for the thorough review. The VEGF acronym has been corrected and explained in full, page 9 line 584-585 as were the other acronyms with careful double checking of the entire manuscript.   

  • Line 326: “Despite of this” should be replaced with “Despite this” or “in spite of this”

This has been corrected.

  • Line 343: microRNAs in circulating exosomes have not been evaluated in this study. Please clarify that exosomes are another source of microRNAs, which is quite different from evaluating them in serum

We have clarified in the manuscript that the content of exosomes were not evaluated in the current study page 11.

  • Line 373: a possible limit of this study is the fact that only obese women were included. Although obesity is a major risk factor for GDM, there are several other risk factors that clearly contribute to the development of this condition, which, in fact, develops even in women with normal BMI. Further studies should evaluate the possible role of these microRNAs as predictors of GDM even women with normal BMI

We agree with the reviewer that a limitation of the current study is the fact that only obese women were included. We have elaborated our limitation statement on page 12.

Round 2

Reviewer 1 Report

The manuscript has been improved. However, one small detail is that the authors missed the words of "Supplementary Figure 1" in the main text. They should insert it in the proper places in the text.